# A Dual Pharmacological Strategy against COVID-19: The Therapeutic Potential of Metformin and Atorvastatin

**DOI:** 10.3390/microorganisms12020383

**Published:** 2024-02-13

**Authors:** Luis Adrián De Jesús-González, Rosa María del Ángel, Selvin Noé Palacios-Rápalo, Carlos Daniel Cordero-Rivera, Adrián Rodríguez-Carlos, Juan Valentin Trujillo-Paez, Carlos Noe Farfan-Morales, Juan Fidel Osuna-Ramos, José Manuel Reyes-Ruiz, Bruno Rivas-Santiago, Moisés León-Juárez, Ana Cristina García-Herrera, Adriana Clara Ramos-Cortes, Erika Alejandra López-Gándara, Estefanía Martínez-Rodríguez

**Affiliations:** 1Unidad de Investigación Biomédica de Zacatecas, Instituto Mexicano del Seguro Social, Zacatecas 98000, Mexico; rdz.carlos09@hotmail.com (A.R.-C.); taneiro87@hotmail.com (J.V.T.-P.); rondo_vm@yahoo.com (B.R.-S.); ana.garciaher@imss.gob.mx (A.C.G.-H.); ramcor16@hotmail.com (A.C.R.-C.); erykagandara@gmail.com (E.A.L.-G.); estefania_940@yahoo.com.mx (E.M.-R.); 2Department of Infectomics and Molecular Pathogenesis, Center for Research and Advanced Studies (CINVESTAV-IPN), Mexico City 07360, Mexico; selvin.palacios@cinvestav.mx (S.N.P.-R.); carlos.cordero@cinvestav.mx (C.D.C.-R.); 3Departamento de Ciencias Naturales, Universidad Autónoma Metropolitana (UAM), Unidad Cuajimalpa, Ciudad de México 05348, Mexico; carlos.farfan@cinvestav.mx; 4Facultad de Medicina, Universidad Autónoma de Sinaloa, Culiacán, 80019, Mexico; osunajuanfidel.fm@uas.edu.mx; 5División de Investigación en Salud, Unidad Médica de Alta Especialidad, Hospital de Especialidades No. 14, Centro Médico Nacional “Adolfo Ruiz Cortines”, Instituto Mexicano del Seguro Social (IMSS), Veracruz 91897, Mexico; jose.reyesr@imss.gob.mx; 6Facultad de Medicina, Región Veracruz, Universidad Veracruzana (UV), Veracruz 91700, Mexico; 7Laboratorio de Virología Perinatal y Diseño Molecular de Antígenos y Biomarcadores, Departamento de Inmunobioquímica, Instituto Nacional de Perinatología, Ciudad de México 11000, Mexico; moisesleoninper@gmail.com

**Keywords:** COVID-19, metformin, atorvastatin, antiviral drugs, pharmacological repositioning

## Abstract

Metformin (MET) and atorvastatin (ATO) are promising treatments for COVID-19. This review explores the potential of MET and ATO, commonly prescribed for diabetes and dyslipidemia, respectively, as versatile medicines against SARS-CoV-2. Due to their immunomodulatory and antiviral capabilities, as well as their cost-effectiveness and ubiquitous availability, they are highly suitable options for treating the virus. MET’s effect extends beyond managing blood sugar, impacting pathways that can potentially decrease the severity and fatality rates linked with COVID-19. It can partially block mitochondrial complex I and stimulate AMPK, which indicates that it can be used more widely in managing viral infections. ATO, however, impacts cholesterol metabolism, a crucial element of the viral replicative cycle, and demonstrates anti-inflammatory characteristics that could modulate intense immune reactions in individuals with COVID-19. Retrospective investigations and clinical trials show decreased hospitalizations, severity, and mortality rates in patients receiving these medications. Nevertheless, the journey from observing something to applying it in a therapeutic setting is intricate, and the inherent diversity of the data necessitates carefully executed, forward-looking clinical trials. This review highlights the requirement for efficacious, easily obtainable, and secure COVID-19 therapeutics and identifies MET and ATO as promising treatments in this worldwide health emergency.

## 1. Introduction

SARS-CoV-2, the etiological agent of COVID-19 disease, has generated an unprecedented epidemic, putting enormous strain on healthcare systems around the world. Despite the widespread use of multiple vaccines, the recurrence of symptomatic cases underscores the necessity of practical treatment approaches against this virus [1,2]. Although vaccines are the primary prevention method, the appearance of viral variants and the risk of inadequate immune responses in some people underline the need for effective therapy [3]. The Food and Drug Administration (FDA) of the United States has approved and permitted the emergency use of medications to treat COVID-19. These cover many therapeutic techniques, from direct antiviral medicines to immunomodulatory [1,4].

On the other hand, the genetic diversity of SARS-CoV-2 is a decisive factor in its infective capacity and transmission dynamics in the human population. Because its RNA polymerase lacks proofreading mechanisms during replication, it has an inherent propensity to accumulate mutations as a single-stranded RNA virus. When these mutations occur in essential areas, such as the S protein, they are critical because they play a role in the virion entry into host cells, affecting the primary target of immune responses and antiviral medicines [5].

Reevaluating current medications for new therapeutic applications has emerged as a critical tactic in this setting. This review focuses on repositioning metformin (MET) and atorvastatin (ATO) as prospective COVID-19 therapy candidates. Traditionally used to treat diabetes [6,7] and dyslipidemia [8,9], these medicines have been demonstrated to have immunomodulatory and antiviral effects that may be useful in treating SARS-CoV-2 infection [10,11,12,13,14,15,16].

It is necessary to find treatments that are safe, effective, and easy to obtain. With their well-known security features and wide availability, MET and ATO offer a unique chance. This review carefully examines the pharmacological basis of both drugs and the latest information on COVID-19 and its antiviral properties.

## 2. Fundamentals and Mechanisms: An In-Depth Analysis of the Pharmacology of Metformin and Atorvastatin

MET (N, N-dimethylbiguanide) is one of the most prescribed drugs worldwide. It is used for treating diabetes, prediabetes, gestational diabetes, polycystic ovarian syndrome, cancer, HIV-associated diabetes, and cardiac ischemia [17]. During the 1920s, guanidine, the active component of Galega officinalis, was used to synthesize multiple antidiabetic compounds, including the main biguanides, MET, and phenformin [18]. Still, MET distinguishes itself due to its favorable profile in treating type 2 diabetes. The structure of biguanides consists of two guanidine groups linked by a methylene bridge (-CH2-). MET has demonstrated a higher safety margin concerning severe complications such as lactic acidosis. Its therapeutic range in hepatic concentrations varies between 10 and 40 µM in type 2 diabetic patients [19]. MET lowers the basal and postprandial plasma glucose (PPG) as a biguanide agent [20]. MET actions ameliorate insulin resistance by inhibiting principally hepatic gluconeogenesis, reducing intestinal glucose adsorption, and improving glucose uptake and utilization. Within the liver, the principal effect of MET is a reduction in the hepatic glucose output. Besides lowering blood glucose levels, MET has additional health benefits, including weight reduction, reducing plasma lipid levels, and preventing some microvascular complications [21]. The drug is not metabolized and is widely distributed into the body tissues, including the intestine, liver, and kidney, by organic cation transporters [22].

MET’s antidiabetic effect is due to the partial reduction of the activity of the respiratory chain on mitochondrial complex I through interfering with electron transfer and ATP production [6,7]. Interestingly, MET inhibits the mitochondrial glycerophosphate (mGPD) activity and mitochondrial respiration from glycerol-3-phosphate at lower concentrations than those affecting complex I [23]. This mechanism leads to other downstream events, including the activation of adenosine monophosphate kinase (AMPK), inhibition of 1,6-bisphosphatase, and glucagon signaling [24].

The defined mechanism of MET action is AMPK activation, inhibiting the liver’s critical enzymes involved in gluconeogenesis and glycogen synthesis while stimulating insulin signaling and glucose transport in muscles [24]. AMPK regulates the cellular and organ metabolism, where any decrease in hepatic energy leads to the activation of AMPK by increasing the adenosine monophosphate (AMP) to adenosine triphosphate (ATP) AMP:ATP and adenine diphosphate (ADP) ADP:ATP ratios, as regulated by adenylate cyclase. AMP binds to the sensing domain, where it allosterically activates AMPK. Also, AMP causes the inhibition of the dephosphorylation of Thr172 on the γ subunits [25].

AMPK activation requires the phosphorylation of Thr172 of the catalytic α-subunit, which is regulated upstream by the tumor suppressor serine/threonine kinase 11 (STK11/LKB1) [26] and by the Ca2+ calmodulin-dependent kinase kinases CaMKK-α and -β [27]. The LKB1 pathway regulates downstream the phosphorylation of the transducer of regulated CREB activity 2 (TORC2) [28]. The activated AMPK cascade is a metabolic switch sensor in cells that switches from an anabolic to a catabolic state, switching off the ATP-consuming synthetic pathways and restoring the energy balance by switching on the uptake of glucose and fatty acids [29].

Similar to all medicines, MET can cause side effects, including nausea, vomiting, diarrhea, indigestion, heartburn, flatulence, weakness, myalgia, chest discomfort, palpitations, flushing, headache, lightheadedness, dyspnea, a flu-like condition, reduced vitamin B12 levels, increased diaphoresis and cutaneous side effects [30]. MET is contraindicated in patients with severe renal failure as there is an increased risk of lactic acidosis [31]. 

On the other hand, ATO is a statin that inhibits 3-hydroxy-3-methyl-glutaryl-CoA (HMG-CoA) reductase (HMGCR). In 1980, it was shown that mevastatin, formerly called ML-236B or compactin, markedly lowers the levels of low-density lipoprotein (LDL) cholesterol [32] and apolipoprotein B (apo-B). ATO is a synthetic variant of this statin, approved in 1996 [8].

ATO exhibits a potent competitive inhibitory effect on the activity of the enzyme HMGCR, which plays a pivotal role in the biosynthesis of cholesterol in the liver. By inhibiting this enzyme, ATO decreases the production of cholesterol. HMGCR inhibition has a rate-limiting effect in the cholesterol synthesis pathway through converting 3-hydroxy-3-methyl glutaryl-CoA (HMG-CoA) into mevalonate, the precursor of cholesterol and other sterols [33]. Consequently, hepatocytes lead to increased production (upregulation) of LDL receptors on their surface, enhancing the uptake and clearance of LDL cholesterol from the bloodstream [8,9].

This reduction in the LDL cholesterol levels helps mitigate the risk of cardiovascular diseases associated with high cholesterol. ATV can also raise the HDL cholesterol levels and decrease triglycerides [18]. The common side effects of ATV happen in more than 1 in 100 people, including gastrointestinal, nausea, bloating, diarrhea or constipation, hepatitis, myalgia, and myopathy [19].

This reduction in the LDL cholesterol levels helps mitigate the risk of cardiovascular diseases associated with high cholesterol. ATO can also raise the HDL cholesterol levels and decrease triglycerides [34]. The common side effects of ATO happen in more than 1 in 100 people, including gastrointestinal, nausea, bloating, diarrhea or constipation, hepatitis, myalgia, and myopathy [35]. 

## 3. Antiviral Properties of Metformin and Atorvastatin

MET and ATO are hypoglycemic and hypolipidemic drugs [8,23], respectively. MET interferes with various stages of the viral life cycle (Table 1). Furthermore, MET impacts host cell metabolism, and the immune response might contribute to its antiviral effects. In addition, the drugs’ hypolipidemic effects may contribute to many viruses requiring cholesterol or other lipids for an efficient replicative cycle (Table 2) [36]. In vitro and in vivo studies show that both drugs efficiently inhibit the infection of a diverse group of viruses such as IAV, HCV, HBV, HPV, HSV, HIV, Rotavirus, and KSHV [37,38,39,40], as well as some RNA viruses such as DENV, ZIKV, YFV [41], including SARS-CoV-2 [42].

The reason why MET and ATO are antiviral drugs is related to lipid metabolism [8,24]. Like SARS-CoV-2 and other RNA and DNA viruses, the viral cycle depends on lipid metabolism, especially on molecules like cholesterol and the formation of lipid droplets, which are essential for the replication cycles of many viruses [52]. 

For example, during the infection of different RNA viruses, such as SARS-CoV-2, lipids are necessary during the replicative cycle. Upon viral entry, lipid bilayers of the virus envelope obtained from the endoplasmic reticulum (ER) membrane participate in viral attachment and fusion. Second, an increase in cholesterol and fatty acid synthesis leads to the formation of invaginations of the ER membrane called replicative complexes (RCs), where viral translation and replication occur. In the next step, a combination of cholesterol-rich RCs used as a scaffold and the accumulation of lipid droplets (LDs) serve as the packaging for the viral genome and the formation of the nucleocapsid and contribute to the assembly of the progeny. Finally, the nucleocapsid buds through the ER membrane, completing the assembly of the virions. Virions are transported through the exocytic pathway to the Golgi complex for maturation and release from the infected cell [41,52,53,54].

The molecular mechanism of action of MET remains partly unknown. However, it has been suggested that, being a cation, it accumulates in the mitochondria due to the electrical gradient of the internal membrane, inhibiting complex I of the mitochondrial respiratory chain [55,56]. Therefore, MET inhibits mitochondrial ATP synthesis and consequently causes the indirect activation of AMPK, which is sensitive to ATP depletion, affecting fatty acid synthesis. Furthermore, MET directly reduces the synthesis of cholesterol and fatty acids (enzymatic inactivation) through the SREBP pathway, involving the lipid requirements of the virus. On the other hand, MET also induces the interferon-mediated response through AMPK [41,52,53].

Mainly, the antiviral mechanism of MET has been associated with the activation of the AMPK protein [24], which is attenuated in the early stages of infection by some viruses, such as DENV [57], promoting a decrease in intracellular lipids [23]. On the other hand, MET also inhibits the replication of DNA viruses, such as hepatitis B virus (HBV) [37], through the repression of genes related to viral transcription [58]. Likewise, MET with Entecavir (a guanosine nucleoside analog) inhibited HBV replication more significantly than the treatments alone [40,58]. Thus, MET promises to be a potential antiviral therapy, especially with other drugs [52]. 

**Table 2 microorganisms-12-00383-t002:** ATO is a broad-spectrum antiviral because it inhibits the cycle replicative of different viruses.

Group Baltimore (Class)	Virus (Family)	Model	Antiviral Effect
IV (ssRNA+)	SARS-CoV-2	In silico, in vitro, and patients	Molecular docking analyses reveal that ATO could bind to the virus’s Mpro protease, Spike protein, and RNA-dependent RNA polymerase [59,60,61]. ATO showed antiviral activity against the D614G, Delta, and Mu variants of SARS-CoV-2 in Vero E6 cells through pre-post, pre-infection, and post-infection treatments [61,62,63].
	DENV	In vivo and in vitro	ATO reduced flaviviruses’ viral titer and cytopathic effect in Huh-7, Vero, MDCK, and neural stem/progenitor cells. ATO reduced clinical signs and increased survival of AG129 mice [53,64,65,66,67,68]. ATO inhibits nuclear-cytoplasmic transport of viral proteins, inhibiting DENV viral replication [65].
	ZIKV		
	YFV		
	HCV	In vitro, and Patients	ATO suppresses HCV replication and has synergistic action with interferon. Additionally, it is associated with a 49% reduction in the incidence of hepatocellular carcinoma [69,70,71,72].
	CV	In vivo	ATO reduced pathological features in the myocardium of mice infected with CVB3m and inhibited viral replication [73].
V (ssRNA-)	IAV	In vitro	ATO inhibits the formation of lipid droplets by IAV, suppressing virus replication [74,75].
	RABV	In vitro	ATO inhibits the formation of lipid droplets by RABV, suppressing virus replication [76].
VI (ssRNA-RT)	HIV	Patients	ATO is safe in patients with HIV, reduces virus-associated inflammation, reduces the activation of CD8^+^ and CD4^+^ T cells, and prevents viral rebound [75,76,77,78,79,80,81] [77,78,79,80,81,82,83].

Recently, it has been shown that AMPK stabilizes ACE2 receptor expression upon phosphorylation of Ser360 [84]. It has been reported that phosphorylation of the ACE2 receptor promotes conformational and functional changes that would prevent SARS-CoV-2 from binding to host cells [84]. Furthermore, it is well known that ACE2 has an essential role in the anti-inflammatory immune response [85]. However, the entry of SARS-CoV-2 through interaction with ACE2 downregulates its expression, causing a pro-inflammatory effect [86,87]. Therefore, these data suggest that MET is an excellent therapeutic strategy against COVID-19, not only because it inhibits virus replication and entry but also because it could decrease the severe effects of the disease (Figure 1).

Interestingly, ATO is another drug with antiviral properties that inhibits lipid synthesis, affecting viral replication, as does MET [8,62,74]. However, studies in cancer cells have revealed that inhibition of lipid synthesis affects the prenylation of small GTPase proteins involved in actin cytoskeleton remodeling, affecting metastasis and intracellular transport [88]. Additionally, statins, including atorvastatin, have affected various cellular pathways during cellular transformation, such as in lung cancer. They observed its effect on survival pathways, pro-apoptotic signaling, chemotactic control, and angiogenesis. Therefore, atorvastatin could play a role in treating this type of cancer [89]. Due to this, Segatori et al., 2021 studied the effect of ATO treatment on nuclear import, demonstrating that ATO can reduce the nuclear localization of nuclear transport receptor proteins such as karyopherin alpha 2 (KPNA2) [63]. On the other hand, they found that the combination of ATO with ivermectin (an antiparasitic drug that blocks importin α/β pathway-dependent nuclear import [90]) more significantly inhibits the nuclear import of KPNA2 [63], suggesting that ATO in combination with ivermectin could be used as an antiviral treatment against viruses that import proteins into the nucleus, such as flaviviruses, influenza viruses and SARS-CoV-2.

Particularly, ATO can block the nuclear import of Huh7 cells, promoting the cytoplasmic accumulation of KPNA1 and KPNA2, as well as a nuclear import dependent on these pathways [65]. Moreover, ATO treatment can block the nuclear import of NS5 polymerase and NS3 viral protease of DENV-2. Importantly, in this work, we report that combining ATO with ivermectin significantly protects male AG129 mice from Dengue disease compared to a single ATO treatment, in addition to reducing the nuclear localization of NS3 in mouse brain tissue cells [65]. We suggest that the pleiotropic effects of ATO may affect the establishment of infection and viral replication.

On the other hand, we have tested combinations of ATO/ezetimibe and MET/ATO (unpublished data) in in vitro and in vivo models. These combinations show better antiviral capacity against viruses, for example, DENV, ZIKV, and YFV infections, than using a single drug alone [53]. Therefore, these drug combinations would be interesting to test in patients. Together, these results strongly suggest that MET and ATO should be considered possible treatments for COVID-19. On the other hand, we have tried combinations of ATO/ezetimibe [10], ivermectin/ATO [17], and MET/ATO (unpublished data) in in vitro and in vivo models. These combinations show better antiviral capacity against viruses, for example, DENV, ZIKV, and YFV infections, than using a single drug alone. Therefore, this drug is a combination that would be interesting to test in patients. Together, these results strongly suggest that MET and ATO should be considered possible treatments for COVID-19.

## 4. The Impact of Metformin and Atorvastatin on the Immune Response

The term “Immunomodulators” pertains to drugs that modify the immune system’s response. These essential medications combat various conditions, such as infectious diseases, tumors, and primary or secondary immunodeficiency. However, these molecules may have differential impacts involved in the given pathology. Some reports have shown that immune modulators can decrease the morbidity and mortality in severely affected patients during viral infections [91]. The antiviral activity of MET is due, in part, to its immunostimulatory effects [92]. These effects mainly depend on AMP-activated protein kinase (AMPK) activation, as described above. Briefly, the activation of MAPK increases mTOR signaling by phosphorylating TSC2 and RAPTOR, a crucial connection between immune function and metabolism [93].

Additionally, this response can inhibit the NF-κB pathway, a master regulator of the inflammatory response [94]. For instance, several studies have demonstrated in animal models that MET decreases inflammatory cytokines, such as IL-1β, IL-6, and TNFα, and increases anti-inflammatory via IL-10 expression [92,93,95]. Thus, it protects against acute lung injuries or can suppress the cytokine storm produced by severe COVID-19 [96]. 

On the other hand, MET modulates the differentiation and activation effects of various immune cells. Macrophages exhibit two primary phenotypes: pro-inflammatory (M1) and “alternatively” activated (M2) macrophages. These distinct macrophage phenotypes are primarily pro-inflammatory responses or the resolution of inflammation [97]. In vitro, MET can have anti-inflammatory preferences. Thus, the response in macrophages stimulated with LPS decreases the TNF-α, IL-6, MCP-1, ROS, and pro-IL-1β while boosting IL-10 expression [98]. However, MET in a cancer environment induces a shift from M2 to M1 polarization of macrophages [99]. This provides evidence of MET’s ability to enhance the balance between M1 and M2 macrophages. In addition, MET has demonstrated a reduction in the neutrophil count in patients with diabetes and polycystic ovarian disease. Furthermore, MET modulates the function of neutrophils by decreasing the formation of neutrophil extracellular traps (NETs) [100]. This effect is particularly relevant considering the involvement of neutrophils in various complications of SARS-CoV-2 infection [101]. 

The adaptive immune response begins with the recognition of specific antigens, marking the commencement of a cascade that transforms naive T cells into effector T cells. These effector cells can differentiate into T-helper cells (CD4) or cytotoxic T cells (CD8). In the MET-treated mouse model, the T-helper response showed a decrease in the levels of Th1 and Th17 cytokines profile through the downregulation of T-box and RORγt transcription factors [102]. Also, upregulation of the anti-inflammatory or immunosuppressive response has been reported, characterized by increased Th2 and T-regulatory cell subpopulations, respectively. Moreover, some reports have demonstrated that MET reduced the levels of autoantibodies in relevant autoimmune disease models. Similarly, MET has shown favorable outcomes in the context of cancer, such as reduced tumor progression and, in some cases, complete tumor eradication. This beneficial effect is attributed to enhancing the downregulation of the immunosuppressive T-regulatory cells and the upregulation of T memory cells [103,104]. 

Although the immunomodulatory effect of ATO has been less studied than that of MET, statins appear to play a crucial role in modulating the immune response at various levels. The growing body of evidence has identified statins as potential anti-inflammatory agents. As described above, several cholesterol metabolites and their associated nuclear receptors can regulate the immune system.

The in vitro studies investigating the impact of ATO on inflammatory cytokines, such as IL-6 and IL-1β, suggest a potential role in reducing their concentrations [105]. If applicable in vivo, this could imply that statins, like ATO, may possess the ability to mitigate the pro-inflammatory effects of these cytokines within tissues. For instance, the use of ATO in Multiple Sclerosis increased the levels of IL-4, IL-10, and TGF-β, while IL-17 and TNF-α decreased compared to the control group [106]. Similarly, in Kawasaki disease, ATO exhibited inhibitory effects in producing soluble mediators of inflammation, including IL-2 and TNF-α, impacting T cell proliferation [107]. 

Additional studies have highlighted the substantial immune-suppressive effects of statins on adaptive immune cells, leading to a decrease in the T cell quantity, impairment of antigen-presenting cell activation of T cells, and a shift toward Th2 cytokines, potentially resolving autoimmunity. Moreover, some reports have demonstrated an influence on humoral immunity in healthy individuals through the short-term statin administration, potentially leading to improved vaccine efficacy and immunomodulatory actions. This warrants further investigation to fully understand their implications in various health conditions [108,109,110].

## 5. Retrospective Studies and Clinical Trials That Have Evaluated the Potential for Use of Metformin and Atorvastatin during COVID-19

Several retrospective studies have examined the efficacy of specific drugs, such as MET and ATO, in combating COVID-19. These findings indicate that MET, a medication routinely recommended for diabetes, may have additional beneficial effects outside its typical application. Notably, diabetic individuals who were already undergoing MET therapy when diagnosed with COVID-19 showed a propensity for improved clinical results. Based on these investigations, it was found that these people had a diminished likelihood of developing severe manifestations of COVID-19. Additionally, they saw a decrease in the necessity for admission to intensive care units due to progression and severity and a lower mortality rate in comparison to individuals who were not using the medication. These findings are noteworthy as they indicate that MET can be used not only as a treatment for diabetes but also as a potential aid in mitigating the severity of COVID-19 [12,111,112,113].

On the other hand, dysbiosis in the intestinal microbiota has been described as contributing to the appearance and progression of many viral diseases, including COVID-19 and its subsequent manifestations (long COVID) [114]. MET has been shown to modulate the intestinal microbiota in patients with COVID-19, increasing the alpha diversity of bacteria, which contributes to the modulation of the immune response and reduces the appearance of long COVID cases [14,115,116].

However, research has also been conducted on ATO, a statin recognized for its role in cholesterol regulation, concerning COVID-19. Several retrospective cohort studies have indicated that this medication may be linked to a reduction in both COVID-19 mortality and progression. Despite the encouraging nature of these findings, they must be interpreted critically on account of the inherent limitations of these research studies [117,118,119]. In addition, it has been described that it can modulate the immune response of patients with COVID-19, preventing the appearance of long COVID, and as a treatment for this [120,121].

However, it is essential to consider these findings within the context of the methodological limitations of the studies and the need for further research, preferably through prospective clinical trials, to confirm these effects and better understand how these drugs can be used effectively against COVID-19.

### 5.1. Clinical Trials with Metformin

Several clinical trials have been conducted to investigate MET usage during COVID-19. A summary of them and their main findings are presented in Table 3.

Evidence suggests that MET may be clinically beneficial for patients with COVID-19. For instance, Reis et al., 2022 conducted a randomized, placebo-controlled clinical trial in Brazil to determine the possible use of MET during COVID-19. The researchers enrolled 421 adults during different waves of SARS-CoV-2 variants to evaluate whether extended-release MET at a dose of 750 mg twice daily would provide benefits over placebo, even in patients already taking 1000 mg of immediate-release MET for conditions such as diabetes, prediabetes, weight loss, polycystic ovary syndrome, or nonalcoholic fatty liver disease. The relative risk of hospitalization or prolonged emergency service visit with MET was 1.03 (95% Bayesian credible interval, 0.64 to 1.66). In this trial, patients were started on MET at 1500 mg daily without dose adjustment, which may have caused side effects and discontinuations. However, hospitalization occurred in 8 of 168 patients (4.8%) in the MET group and 14 of 179 patients (7.8%) in the control group [10]. The authors suggest that MET was crucial in early treatment during the health crisis caused by COVID-19. 

Another study was carried out by Ventura-López et al., 2022. The authors claim that MET is a therapeutic option for COVID-19 patients. They first evaluated the impact of MET in an in vitro model using lung cells infected with SARS-CoV-2 clinically isolated (alpha, delta, and epsilon variants of SARS-CoV-2). In the in vitro aspect, MET effectively inhibited viral replication after 48 h of exposure (IC50 189.8 µM) without showing toxicity up to doses of 100 µM and was effective against the variants studied [11].

The clinical trial in patients with SARS-CoV-2 infection and diagnosis of type 2 diabetes mellitus, performed as an adaptive, randomized, prospective, longitudinal, double-blind, multicenter, phase IIb study, compared MET administered at 620 mg twice daily for 14 days vs. placebo. In patients treated with MET, changes were observed in the levels of AST (33.8 to 31.5 U/L), lymphocytes (8.1 to 17%), neutrophils (85.2 to 76.2%), D-dimer (480 to 634 ng/mL), CPR (4.16 to 0.10 mg/L), DHL (257.5 to 256.5 µL/L) and IgG (969 to 781 mg/dL) from the beginning to the end of the study. Additionally, the MET group showed a notable reduction in the viral load, decreasing by 93.2% in just 3.3 days, compared to a 78.3% reduction in 5.6 days in the placebo group. The average hospitalization time was shorter in the MET group (8.8 days) compared to the placebo group (9.8 days), and the patients treated with MET required less supplemental oxygen (5.9 vs. 10.6 points) [11]. The average hospitalization time was shorter in the MET group (8.8 days) compared to the placebo group (9.8 days), and the patients treated with MET required less supplemental oxygen (5.9 vs. 10.6 points) [11]. Regarding safety, mild adverse events were recorded in both groups, with no incidences of hypoglycemia. This study highlights the potential efficacy of MET in decreasing the SARS-CoV-2 viral load and improving the clinical outcomes in patients with COVID-19 and type 2 diabetes mellitus [11].

Clinical studies were conducted by Bramante et al. in 2022, with a phase 3, double-blind, randomized, and placebo-controlled design. They tested the effectiveness of three repurposed drugs (MET, ivermectin, and fluvoxamine) in 1431 US patients with COVID-19 to prevent severe disease cases in non-hospitalized adults. The patients were between 30 and 85 years old, and all were overweight or obese. The outcome variables evaluated were hypoxia (≤93% oxygen saturation on home oximetry), emergency department visits, hospitalization, or death. The groups received the trial drugs according to the following doses: immediate-release MET administered with a dose escalation over six days to 1500 mg per day for 14 days, ivermectin at an amount of 390 to 470 μg per kilogram per day for three days, and fluvoxamine at a dose of 50 mg twice a day for 14 days [13].

Additionally, patients were asked to record the severity of their daily symptoms in paper diaries over 14 days. Neither overall signs nor specific symptoms of COVID-19 were reduced more rapidly with placebo than with any of the trial drugs. This study highlights no serious adverse events related to the medication [13]. 

Ultimately, the clinical trial concluded that none of the medications were effective in preventing hypoxia, ICU visits, hospitalization, or death. However, the authors mention that in the case of MET, there was a trend of benefit for the prevention of the severe form, according to their criteria (visit to the emergency department, hospitalization, or death), so they argue that more studies are needed with different doses or other populations to determine if any of the antiviral mechanisms (reduction of hepatitis C viral load [123]) or anti-inflammatory have clinical activity in the treatment of COVID-19 [13].

Derived from the previous study, two more studies were generated. In one of them, the authors followed up for up to 300 days with 1126 participants from the original study, who received MET plus ivermectin (group 1), MET plus fluvoxamine (group 2), MET plus placebo (group 3), ivermectin plus placebo (group 4), fluvoxamine plus placebo (group 5) or placebo plus placebo (group 6)), and they found that MET reduced the incidence of post-COVID-19 (4.1% absolute reduction) compared to the placebo. The main post-COVID-19 symptoms were fatigue, difficulty focusing, difficulty sleeping, difficulty breathing, headache, loss of taste, and depression [14].

The latest work was presented during a conference on retroviruses and opportunistic diseases in the USA by Boulware et al., 2023. In this work, they mention that MET has in vitro activity against SARS-CoV-2. Furthermore, they mention that in a randomized phase 3, quadruple-blind, placebo-controlled trial in 1323 participants, MET (1000 mg/day on days 2 to 5; 1500 mg/day on days 6 to 14) resulted in a 42% reduction in ER visits, hospitalizations, and deaths on day 14, a 58% reduction in hospitalizations and deaths on day 28, and a 42% reduction in long COVID in 10 months [122].

Additionally, they mention that MET reduced the viral load by 4.4 times compared to the placebo from baseline to follow-up (5, 10, and 14 days). The antiviral effect increased with an increasing MET dose on days 6 to 14. Conversely, they mention that the antiviral effect was more significant in the unvaccinated group (mean −0.95 log copies/mL) than in the vaccinated group (mean −0.39 log copies/mL). There were no changes in the viral load versus the placebo for ivermectin or fluvoxamine [122].

Finally, the authors conclude that MET reduced the viral load of SARS-CoV-2 in this clinical trial. The temporal relationship with dose titration suggests a dose-dependent effect. The magnitude of the antiviral effect was similar to that of nirmatrelvir on day five and more significant than that of nirmatrelvir on day ten. And they emphasize that MET is safe, widely available, and has few contraindications [122].

### 5.2. Clinical Trials with Atorvastatin

Several clinical trials investigated ATO usage during COVID-19. The following is a summary of them and their principal findings (Table 4).

On the other hand, ATO, a statin used to reduce cholesterol levels, has been researched due to its anti-inflammatory and anticoagulant properties. Davoodi et al., 2021 conducted a randomized controlled clinical trial on 40 Iranian adults hospitalized with COVID-19. Patients were randomly assigned (1:1) to a treatment group receiving ATO 40 mg + lopinavir/ritonavir 400/100 mg or a control group receiving lopinavir/ritonavir alone. ATO was administered as one tablet daily. In the case of lopinavir/ritonavir, it was administered twice daily. Both treatments were administered for five days [15].

The primary outcome of the trial was the length of hospitalization; the secondary outcomes were the need for interferon or immunoglobulin, receipt of invasive mechanical ventilation, O2 saturation (O2 sat), and C-reactive protein (CRP) level, assessed at baseline and on the sixth day of treatment [15].

The CRP level decreased significantly in the lopinavir/ritonavir + ATO group (*p* < 0.0001, Cohen’s d = 0.865) so that there was a significant difference in the CRP level on the sixth day between the two groups (*p* = 0.01). However, there was no significant difference in O2 sat on day 6. Although the duration of hospitalization in the lopinavir/ritonavir + ATO group was significantly reduced compared with the control group (*p* = 0.012), there was no significant difference in the invasive mechanical ventilation, receipt, and need for interferon and immunoglobulin. The authors conclude that combining ATO + lopinavir/ritonavir may be more effective than lopinavir/ritonavir in treating adult patients hospitalized with COVID-19 [15].

In another randomized controlled trial published in *The BMJ*, in 11 hospitals in Iran, 605 patients >18 years of age with COVID-19 admitted to the ICU participated. ATO 20 mg was administered orally once daily versus placebo; follow-up was performed for 30 days from randomization, regardless of the hospital discharge status [16].

Death occurred in 90 (31%) patients in the ATO group and 103 (35%) in the placebo group (odds ratio 0.84, 95% confidence interval 0.58 to 1.22). The venous thromboembolism rates were 2% (*n* = 6) in the ATO group and 3% (*n* = 9) in the placebo group (odds ratio 0.71, 95% confidence interval: 0.24 to 2.06). Myopathy was not diagnosed clinically in either group. The liver enzyme levels increased (2%) in five patients assigned to ATO and six assigned to placebo (odds ratio 0.85, 95% confidence interval 0.25 to 2.81). The main drugs in the co-treatment were aspirin and antivirals such as remdesivir and favipiravir [16].

The authors conclude that ATO was not associated with a significant reduction in composite venous or arterial thrombosis, extracorporeal membrane oxygenation treatment, or all-cause mortality compared with placebo. However, the treatment was found to be safe. The overall event rates were lower than expected, so a clinically significant treatment effect cannot be excluded. Furthermore, they found a slight reduction in white blood cells, platelets, and D-dimer levels in the patients treated with ATO [16].

Visos-Varela et al., 2023 analyzed the chronic use of statins (simvastatin, lovastatin, pravastatin, fluvastatin, ATO, rosuvastatin, and pitavastatin) and severe outcomes of COVID-19 (risk of hospitalization and mortality), progression to severe consequences and susceptibility to the virus. The study evaluated the risk of hospitalization, mortality, progression, and susceptibility to COVID-19. They collected data on 2821 hospitalized cases, 26,996 non-hospitalized cases, and 52,318 controls [121].

Chronic use of ATO was associated with a decreased risk of hospitalization (adjusted odds ratios [aOR] = 0.83; 95% confidence interval [CI]: 0.74–0.92) and mortality (aOR = 0.70; 95% CI: 0.53–0.93), partly attributable to a lower risk of susceptibility to the virus (aOR = 0.91; 95% CI: 0.86–0.96) [121].

On the other hand, simvastatin was associated with a reduced mortality risk (aOR = 0.59, 95% CI: 0.40–0.87). The large degree of heterogeneity observed in the estimated OR of the different statins suggests that there was no class effect. In conclusion, the authors suggest that chronic use of ATO (and, to a lesser extent, simvastatin) is associated with a decreased risk of severe COVID-19 outcomes [121].6. Conclusions and Future Perspectives

The possibility of MET and ATO as therapeutic therapies for COVID-19 is highly promising. These medications, commonly used for treating diabetes and dyslipidemia, have shown notable immunomodulatory and antiviral effects that could be crucial against SARS-CoV-2. Ensuring effective, widely available, and safe treatments is especially critical during the current worldwide health emergency.

Notably, the cost-effectiveness of these treatments must not be disregarded. MET and ATO, being generic pharmaceuticals, represent a cost-efficient [124,125] substitute for more recent, trademarked anti-SARS-CoV-2 medications like Paxlovid^®^ [126]. Their FDA and Cofepris (Mexico) approvals for diabetes management demonstrate their thorough examinations for safety and effectiveness [127,128], which increases confidence in their use for COVID-19 based on positive results from earlier clinical trials.

The role of MET extends beyond managing diabetes, as its antidiabetic actions can potentially reduce the severity and mortality of COVID-19. Through the partial inhibition of mitochondrial complex I and the activation of AMPK, MET enhances insulin sensitivity and has possible antiviral properties. Moreover, the capacity of ATO to reduce cholesterol levels and regulate immunological responses could be quite beneficial, particularly in severe instances of COVID-19 that are distinguished by an excessively aggressive immune response.

The available evidence, however varied, tends to favor optimism. Several research studies suggest that using MET and ATO can effectively reduce the severity and mortality of COVID-19, while some studies advise interpreting the results with caution. The documented advantages in specific patient cohorts, such as decreased hospitalizations and mortality rates, underscore their potential to impact patient outcomes effectively.

However, the journey from observation to clinical implementation is fraught with obstacles. The presence of diverse study approaches and patient demographics necessitates meticulous data analysis. Conducting prospective randomized controlled trials is crucial for confirming these results and finding the most efficient use of MET and ATO in treating COVID-19. Furthermore, considering the wide range of possible adverse effects, it is necessary to reevaluate these treatments’ widely recognized safety profiles in COVID-19.

Amidst the ongoing challenges posed by the COVID-19 pandemic, repurposing MET and ATO is a promising and innovative gesture. Due to their cost-effectiveness, wide availability, and extensive safety records, they are highly suitable for reapplication. Nevertheless, this endeavor necessitates thorough investigation, precise examination, and a steadfast commitment to comprehending their significance in the COVID-19 storyline. By following this road, we will be able to fully utilize the therapeutic capabilities of MET and ATO, providing a more promising future in this ongoing global health crisis.

## Figures and Tables

**Figure 1 microorganisms-12-00383-f001:**
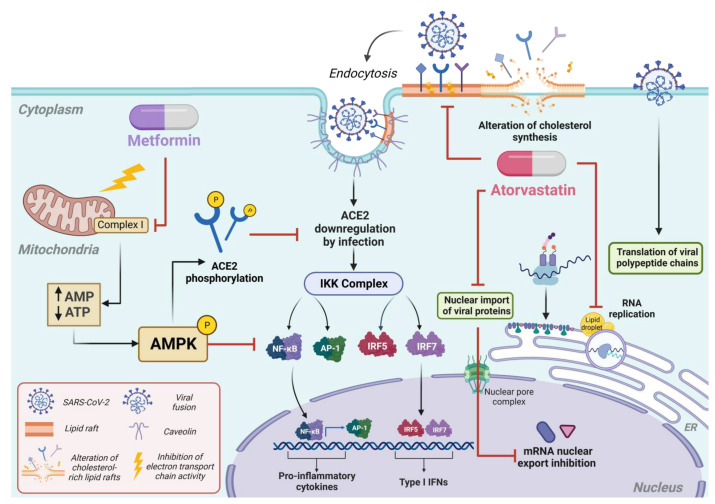
Antiviral capabilities of the MET and ATO drugs. The antiviral mechanism of MET is associated with inhibiting the electron transport chain, which helps the phosphorylation of AMPK and, subsequently, of the ACE2 receptor, preventing the entry of SARS-CoV2. On the other hand, it has been observed that viral entry causes a decrease in ACE2, triggering an exacerbating pro-inflammatory immune response. This allows us to suppose that MET can prevent viral entry and replication and the pro-inflammatory events caused by the disease. Like other statins, ATO has been associated mainly with intracellular lipid depletion, which can affect different viral stages, such as viral entry and replication. One of its effects may be the destabilization of lipid rafts, where viral receptors are anchored, and the decrease in lipids in the endoplasmic reticulum that allows the formation of new viral particles. Also, ATO can prevent the transport of viral proteins to the nucleus by preventing the sequestration of cellular messenger RNA, decreasing viral replication.

**Table 1 microorganisms-12-00383-t001:** MET is a broad-spectrum antiviral because it inhibits the replicative cycle of different viruses.

Group Baltimore (Class)	Virus (Family)	Model	Antiviral Effect
I (dsDNA)	HPV	In vitro, in vivo, and patients	MET promotes HPV-HNSCC and cervical cancer cell apoptosis. It can also increase CD8^+^ Teff and FoxP3^+^ Tregs in the TME, suggesting an immunomodulatory effect [38,43].
	HSV	In vitro	Co-administration of C-REV (attenuated oncolytic HSV-1) and MET produces an antitumor effect and prolonged survival in mice. Additionally, it improves systemic antitumor immunity [44].
	KSHV	In vitro	MET induces apoptosis in primary effusion lymphoma (KSHV-associated aggressive B-cell lymphoma) cells [45].
III (dsRNA)	Rotavirus (Reoviridae)	In vitro and in vivo	MET inhibits the Rotavirus gene and protein expression in Caco-2 cells. Furthermore, MET mitigates intestinal lesions caused by Rotavirus [39].
IV (ssRNA+)	SARS-CoV-2	In vitro and patients	MET effectively inhibits viral replication of SARS-CoV-2 after 48 h of drug exposure without cytotoxic effect in Vero [11], Calu3, and Caco2 cells by up to 99% [42].
	DENV	In vitro and in vivo	MET inhibited ZIKV infection in Huh-7, C20, and U-87 cells but inhibited DENV and YFV infection in Huh-7 cells more effectively. During DENV infection alone, MET increased the survival of male AG129 mice, reducing severe signs of the disease [41,46].
	ZIKV		
	YFV		
	HCV	In vitro	Simvastatin and MET inhibited cell growth and HCV infection in Huh7.5 cells. Furthermore, MET increased cell death markers, activated type I interferon signaling, and inhibited HCV replication through AMPK activation [47,48].
V (ssRNA-)	IAV	In vitro, in vivo, and patients	Studies showed that MET inhibits viral replication and cytokine expression induced by IAV. MET treatment was associated with decreased influenza-related mortality in diabetic patients [49].
VI (ssRNA-RT)	HIV	Patients	Co-administration of antiretrovirals and MET in patients with HIV decreased the infiltration of CD4 + T cells in the colon and the activation/phosphorylation of mTOR. Additionally, decreased HIV-RNA/HIV-DNA ratios, a surrogate marker of viral transcription [50]. Co-administration of dolutegravir and MET in patients with HIV and diabetes improves the control of both conditions, with a reduction in viral load and control of HgbA1C [51].
VII (dsDNA-RT)	HBV	In vitro	MET shows an HBV-associated inhibitory effect by negatively regulating the HULC/p18/miR-200a/ZEB1 signaling pathway [40].

**Table 3 microorganisms-12-00383-t003:** Clinical trials with MET.

Authors	Sample size (Patients)	Dosage	Primary Outcomes
Reis et al., 2022 [10].	421	750 mg (extended-release) twice daily for ten days	Presence of side effects and interruptions. Hospitalization in 8 of 168 patients (4.8%) (MET group) vs. 14 of 179 patients (7.8%) (control group).
Ventura-López et al., 2022 [11].	20	620 mg twice daily for 14 days vs. placebo *	MET significantly reduced viral load (93.2%) in 3.3 days and decreased hospitalization time to 8.8 days and supplemental oxygen requirement. Changes were observed in AST, lymphocytes, neutrophils, D-dimer, CRP, DHL, and IgG biomarkers. The average hospitalization time was 8.8 days for the MET group and 9.8 days for the placebo group. Safety was comparable between groups, with no serious adverse events or hypoglycemia.
Bramante et al., 2022 [13].	1431	They tested three drugs. Immediate-release MET was administered for 1–6 days (750 mg) and 7–14 days (1500 mg per day), ivermectin 390 to 470 μg per kilogram per day for three days, and fluvoxamine at 50 mg twice a day for 14 days.	No adverse events occurred. None of the medications were effective in preventing hypoxia, ICU visits, hospitalization, or death. However, the authors mention that in the case of MET, there was a trend of benefits for preventing the severe form.
Bramante et al., 2023 [14].	1126		MET reduced the incidence of post-COVID-19 (4.1% absolute reduction)
Boulware et al., 2023 [122].	1323	MET 1000 mg/day on days 2 to 5; 1500 mg/day on days 6 to 14 *	42% reduction in ER visits, hospitalizations and deaths by day 14, and 58% reduction in hospitalizations and deaths by day 28 42% reduction in long COVID in 10 months Decrease in viral load compared to placebo

* These doses of MET showed better results in the treatment of COVID-19.

**Table 4 microorganisms-12-00383-t004:** Clinical trials with ATO.

Authors	Sample Size	Dosage	Primary Outcomes
Davoodi et al., 2021 [15].	40	40 mg ATO + 400/100 mg lopinavir/ritonavir (group 1) or 400/100 mg lopinavir/ritonavir for 5 days *	The duration of hospitalization was significantly reduced in the lopinavir/ritonavir + ATO group.
BMJ 2022 [16].	605	20 mg oral ATO once daily and placebo each for 30 days	90 (31%) patients died in the ATO group and 103 (35%) in the placebo group. Venous thromboembolism rates were 2% (*n* = 6) in the ATO group and 3% (*n* = 9) in the placebo group. ATO is safe with few adverse effects. Reduction in levels of white blood cells, platelets, and D-dimer in patients treated with ATO

* These doses of MET showed better results in the treatment of COVID-19.

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
