# Peer review of "A Dual Pharmacological Strategy against COVID-19: The Therapeutic Potential of Metformin and Atorvastatin"

_microorganisms, 2024, doi:10.3390/microorganisms12020383_

Round 1

Reviewer 1 Report

Comments and Suggestions for Authors

Manuscript presents some important data about the pondering of the possibility of repurposing of Atorvastatin including analysis of some clinical studies, however, in the Metformin part, fails trying to present a fairly well-known topic for more or less a decade.

In addition, since this is a review, there are studies with significantly information about the use of metformin in COVID (including some from MDPI editorial) that have been ignored for the authors. Mechanistic hypothesis of how the new effect of the drugs could be ocurring have also been ignored and although the title speaks of a dual effect study, the analysis of both drugs is actually analyzed separately.

Comments on the Quality of English Language

Document needs to be thoroughly reviewed for syntax, gramamr and spelling, in addition to being evaluated by a native English speaker.

Author Response

Reviewer 1

Manuscript presents some important data about the pondering of the possibility of repurposing of Atorvastatin including analysis of some clinical studies, however, in the Metformin part, fails trying to present a fairly well-known topic for more or less a decade.

Reply: Thank you for your comments. We agree that the use of metformin has been extensively studied for many other public health issues. However, the studies we have focused on in the present review are solely centered around the case of COVID. Nevertheless, we have included discussions supporting the use of metformin as a therapeutic approach in the context of COVID.

We discussed this in part 3. Antiviral Properties of Metformin and Atorvastatin as “MET and ATO are hypoglycemic and hypolipidemic drugs [8,23], respectively. MET interferes with various stages of the viral life cycle (table 1). Furthermore, MET has been shown to impact on host cell metabolism and immune response might contribute to its antiviral effects. Besides. The drugs' hypolipidemic effects may contribute to many viruses requiring cholesterol or other lipids for an efficient replicative cycle (table 2) [36]. In vitro and in vivo studies show that both drugs efficiently inhibit the infection of a diverse group of viruses such as IAV, HCV, HBV, HPV, HSV, HIV, Rotavirus, and KSHV [37–40], as well as some RNA viruses such as DENV, ZIKV, YFV [41], including SARS-CoV-2 [42 . “

We discussed this in part 5, some clinical studies.

In addition, since this is a review, there are studies with significantly information about the use of metformin in COVID (including some from MDPI editorial) that have been ignored for the authors.

Reply: Thank you for your comments. We have added more studies throughout the text on the use of MET in COVID-19, including studies published in MDPI.

Mechanistic hypothesis of how the new effect of the drugs could be ocurring have also been ignored and although the title speaks of a dual effect study, the analysis of both drugs is actually analyzed separately.

Reply: Thank you for your comment.

Due to the pleiotropic effects of both drugs, the exact mechanism of the combined effect of these drugs is difficult to propose. Instead, we propose the existence of an antiviral synergy derived from the addition of the pleiotropic effects of each drug individually.  As detailed in section 2 and 3 of the manuscript, statins not only impact the mevalonate pathway, but also exhibit pleiotropic effects by intervening in cellular processes, such as prenylation of small G proteins, production of inflammatory mediators, nucleus-cytoplasmic communication, etc. Similarly, Metformin, beyond its hypoglycemic function, presents pleiotropic effects associated with AMPK protein activation, involved in several signaling pathways related to anabolism, catabolism, immune response, inflammatory processes, and other cellular processes. In this context, the characterization of such an effect could be approached initially through the analysis of viral replication and host inflammatory and immune response in preclinical and clinical trials, and subsequently, the antiviral mechanisms underlying this combination could be investigated.

We have checked the grammar of the manuscript.

Reviewer 2 Report

Comments and Suggestions for Authors

dear Authors,

I send you my comments:

1) Please could you clarify the type of review?

2) I have not understand if you think that ATO and MET could be used to treat COVID-19 in patients without hypercholesterolemia or diabetes.

3) please add the suggested dosage of met and ato in patients with covid-19, but you are sure that this treatment could act in low time?

4) probably ato could be used to reduce the effects of long covid, could you add this in discussion?

5) In molecular mechanism of ato, please add this manuscript doi: 10.3390/ph15050589    

Comments on the Quality of English Language

none

Author Response

Reviewer 2

dear Authors,

I send you my comments:

1) Please could you clarify the type of review?

Reply: Thanks for your suggestions. The review we write is a narrative type.

2) I have not understand if you think that ATO and MET could be used to treat COVID-19 in patients without hypercholesterolemia or diabetes.

Reply: Thanks for your comments. Both clinical trials and retrospective studies have been carried out in patients with diabetes (for MET) (Table 3) and with hypercholesterolemia (for ATO) (Table 4); it would be interesting to carry out clinical trials in patients without this history. However, given the antiviral results observed in other viral diseases (Table 1 and 2) in patients without underlying diseases, the results may hold.

3) please add the suggested dosage of met and ato in patients with covid-19, but you are sure that this treatment could act in low time?

Reply: Thanks for your comments. According to clinical studies, the doses that have shown satisfactory results in the treatment of COVID-19 are those of Ventura-Lopez et al. 2022 who tried 620mg of MET twice a day for 14 days, and the results by Boulware et al. 2023 who tested 1000 mg/day of MET on days 2 to 5 and 1500 mg/day from days 6 to 14 post-infection. These doses have shown fewer side effects, a reduction in viral load, a shorter hospital stay and oxygen requirement, and a reduction in long-COVID cases.

In the case of ATO, the study by Davoodi et al. Doses of 40 mg of ATO for five days were shown to reduce the duration of hospitalization. On the other hand, BMJ 2022 demonstrated that 20 mg/day of ATO for 30 days reduces some severity biomarkers. However, we believe that more prospective studies are needed to find the appropriate doses for the treatment of COVID-19, which demonstrate results similar to those reported by the retrospective studies.

All these data are presented in Tables 3 and 4 and section 5, and in these, we add an asterisk to the doses that have shown the best results.

4) probably ato could be used to reduce the effects of long covid, could you add this in discussion?

Reply: Thanks for your comments. We have added a section on the use of ATO and MET to prevent long-COVID. Line 334-358.

5) In molecular mechanism of ato, please add this manuscript doi: 10.3390/ph15050589   

Reply: Thank you for your observation; we have added a sentence about this study.

Line 215-218: “Additionally, statins, including atorvastatin, have affected various cellular pathways during cellular transformation, such as in lung cancer. Observing its effect on survival pathways, pro-apoptotic signaling, chemotactic control, and angiogenesis. Therefore, atorvastatin could play a role in treating this type of cancer [88].

Reviewer 3 Report

Comments and Suggestions for Authors

The authors of the review paper A Dual Pharmacological Strategy against COVID-19: The Therapeutic Potential of Metformin and Atorvastatin have reviewed the possibility of Metformin (MET) and Atorvastatin (ATO) use, commonly prescribed for diabetes and dyslipidemia, respectively, as versatile medicines against SARS-CoV-2. The review paper is interesting, well structured, with four Tables and one Figure in it. Tables and especially Figure are adding to the clarity of the paper. The authors are concluded that repurposing MET and ATO could be promising, innovative and cost-effective treatment for COVID-19.

Comments:

1.      In the Abstract the authors claim that MET: “can partially block mitochondrial complex I and stimulate AMPK, which indicates that it could be used more widely in managing viral infections.” (row 31, 32).

There should be brief explanation why blocking of mitochondrial complex and stimulation of AMPK have anti-viral effects?

2.      Too many names are on the author’s list. The number of authors should be reconsidered (paper has 15 authors), and some moved in acknowledgement section. Authors should have in mind that this is a review paper, not clinical study. Some of contributions listed do not deserve authorship of the paper.

3.      English is good in the paper, but small adjustments are needed.

Comments on the Quality of English Language

Moderate changes only!

Author Response

Reviewer 3

The authors of the review paper “A Dual Pharmacological Strategy against COVID-19: The Therapeutic Potential of Metformin and Atorvastatin” have reviewed the possibility of Metformin (MET) and Atorvastatin (ATO) use, commonly prescribed for diabetes and dyslipidemia, respectively, as versatile medicines against SARS-CoV-2. The review paper is interesting, well structured, with four Tables and one Figure in it. Tables and especially Figure are adding to the clarity of the paper. The authors are concluded that repurposing MET and ATO could be promising, innovative and cost-effective treatment for COVID-19.

Comments:

  1. In the Abstract the authors claim that MET: “can partially block mitochondrial complex I and stimulate AMPK, which indicates that it could be used more widely in managing viral infections.” (row 31, 32).

There should be brief explanation why blocking of mitochondrial complex and stimulation of AMPK have anti-viral effects?

Reply: Thanks for your comments. We have added the MET antiviral mechanism, line 160-180.

For example, during the infection of different RNA viruses, such as SARS-CoV-2, lipids are necessary during the replicative cycle. Upon viral entry, lipid bilayers of the virus envelope obtained from the endoplasmic reticulum (ER) membrane participate in viral attachment and fusion. Second, an increase in cholesterol and fatty acid synthesis leads to the formation of invaginations of the ER membrane called replicative complexes (RCs), where viral translation and replication occur. In the next step, a combination of cholesterol-rich RCs used as a scaffold and the accumulation of lipid droplets (LDs) serve for the packaging of the viral genome and the formation of the nucleocapsid and contribute to the assembly of the progeny. Finally, the nucleocapsid buds through the ER membrane, completing the assembly of the virions. Virions are transported through the exocytic pathway to the Golgi complex for maturation and release from the infected cell [41,52,53].

The molecular mechanism of action of MET remains partly unknown; However, it has been suggested that being a cation, it accumulates in the mitochondria due to the electrical gradient of the internal membrane, inhibiting complex I of the mitochondrial respiratory chain [54,55]. Therefore, MET inhibits mitochondrial ATP synthesis and consequently causes the indirect activation of AMPK, which is sensitive to ATP depletion, affecting fatty acid synthesis. Furthermore, MET directly reduces the synthesis of cholesterol and fatty acids (enzymatic inactivation) through the SREBP pathway, involving the lipid requirements of the virus. On the other hand, MET also induces the interferon-mediated response through AMPK [41,52,53].

  1. Too many names are on the author’s list. The number of authors should be reconsidered (paper has 15 authors), and some moved in acknowledgement section. Authors should have in mind that this is a review paper, not clinical study. Some of contributions listed do not deserve authorship of the paper.

Reply: Thanks for your comments. All authors should be included in the review since they collaborated on different aspects. We have updated the contributions of each author.

  1. English is good in the paper, but small adjustments are needed.

Reply: Thanks for your comments. We have checked the grammar of the manuscript.